# Effect of Stitching, Stitch Density, Stacking Sequences on Low-Velocity Edge Impact and Compression after Edge Impact (CAEI) Behavior of Stitched CFRP Laminates

**DOI:** 10.3390/ma15248822

**Published:** 2022-12-10

**Authors:** Jiamei Lai, Ze Peng, Zhichao Huang, Meiyan Li, Mingzhi Mo, Bangxiong Liu

**Affiliations:** 1Polymer Processing Research Laboratory, School of Advanced Manufacturing, Nanchang University, Nanchang 330031, China; 2Key Laboratory of Conveyance and Equipment, Ministry of Education, East China Jiaotong University, Nanchang 330013, China

**Keywords:** stitched CFRP laminates, edge-impact, compression after edge impact (CAEI), micro-CT

## Abstract

Low-velocity edge impact and compression after edge impact (CAEI) behavior of stitched carbon-fiber-reinforced plastic (CFRP) laminates were experimentally investigated in the paper. Five groups, including three stacking sequences (P_1_, P_2_, P_3_) and two stitch densities (stitch space × stitch pitch is 10 mm × 10 mm and 15 mm × 15 mm) of stitched/unstitched CFRP laminates, were prepared by the VARTM technique and subjected to low-velocity edge impact and compression after edge impact experiments. The damage of CFRP laminates was detected by optical observation and micro-CT. The effects of stitching, stitch density, stacking sequences and impact energy on properties of edge impact and CAEI were discussed. The results show that the damage of edge impact of stitched laminates is smaller than that of unstitched laminates. The main failure mode of CAEI of the unstitched laminates is delamination and that of the stitched laminates is global buckling. The addition of stitches can effectively improve the edge impact resistance and damage tolerance of CFRP laminates. Compared with the unstitched laminates with the same stacking sequence, the peak impact force of the laminates with stitch density 15 mm × 15 mm increases by 5.61–12.43%, and the increase in residual compression strength is up to 5–20.9%. The peak impact force of the laminates with stitch density 10 mm × 10 mm increases by 8.1–31.4%, and the increase in residual compression strength is up to 24.2–27%. Compared with the other two stacking sequences (P_1_ and P_2_), the stacking sequence P_3_ has excellent resistance of edge impact and CAEI properties.

## 1. Introduction

Carbon-fiber-reinforced plastics (CFRP) have been widely used in various aerospace structures because of their excellent properties such as being lightweight and high-strength [1,2]. However, it is subjected vulnerably to the impact of foreign objects, such as accidental tools dropping in the process of aircraft operation or maintenance, or small stones splashing on the platform during takeoff. The impacts, which come from different angles with different energies, may cause serious damage to the composite aircraft structure and result in a significant reduction in its residual strength [3,4,5], thus leading to a great threat to the flight safety of the aircraft. Therefore, it is urgent to improve CFRP impact resistance and damage tolerance by exploring the damage and impact properties of CFRP under different impact conditions, and the results can provide a reference for the design of aircraft structures.

Stitching is a process that effectively improves the impact resistance of CFRP by adding reinforced fiber sutures along the thickness direction of the laminate, which can improve the interlaminar properties. It is a simple and low-cost process, which is of wide concern by the composite industry and has good application prospects [6,7,8]. However, the performance enhancement is only achieved for the vertical impact on the center of composite panels.

Until now, the research on impact damage characteristics of composites by domestic and foreign scholars mostly focused on the vertical impact on the center of the composite panels [9,10,11,12,13,14,15,16]. However, research on the edge impact properties of composite laminates was rare until 2008, when Malhotra [17] proposed the idea that composites near special locations such as the edges of aircraft wings, inspection ports or the edges of other holes in the aircraft are highly susceptible to edge impact from foreign objects, that scholars gradually began to investigate the edge impact and compression after edge impact performances of composites. According to previous studies, composite laminates may face more severe failure when subjected to edge impact than vertical impact [3,18,19,20,21,22,23,24,25,26,27,28]. Damage and behavior of stiffened composite panels [18,19] and CFRP laminates [20,21] subjected to low-velocity edge impact and CAEI were discussed by using numerical simulation and experimental investigation. Kashiwagi [22] investigated the influence of the damage of edge impact on the strength of the CAEI of CFRP laminates. Ostré [23,24] used finite element and experimental analysis to study the effect of the stacking sequences on properties of edge impact and the CAEI of CFRP laminates. Malhotra [25], Rhead [26] and Liu [27] discussed the effect of impact location on the properties of impact and CAI of composite laminates. Arteiro [28] predicted the damage shape and compression strength of the CAEI of CFRP laminates. Therefore, it becomes critical to understand the damage mechanism of edge impact and try to improve the edge impact properties of laminates.

Based on previous research, the stitching process was introduced to the experimental study of the edge impact and compression after the edge impact of stitched/unstitched composite laminates. The effects of stitching, stitch density, stacking sequence and impact energy were discussed to explore the effects of stitching on the edge impact and the CAEI properties of CFRP laminates.

## 2. Experiment

### 2.1. Material Preparation

The CFRP laminate specimens were prepared in three steps. First, a modified locking stitching method (shown in Figure 1a) was used to make the preform of the stitched CFRP laminate (Figure 1b) according to three selected stacking sequences: P_1_ [−45/0/90/45/90]_2s_, P_2_ [45/90/0/−45/0]_2s_ and P_3_ [45/0/90/0/−45]_2s_. The material used was CF12-300 carbon fiber fabric and Kevlar 29 stitches with two stitch densities of 15 mm × 15 mm and 10 mm × 10 mm (it meant stitch space × stitch pitch is 15 mm × 15 mm and 10 mm × 10 mm), and the diameter of the needle was 2 mm. Second, the stitched/unstitched CFRP laminates were prepared by the vacuum-assisted resin transfer molding (VARTM) process. The matrix material was R688 epoxy resin, and the curing agent was amine H3268; the mixing mass ratio of resin to curing agent was 5:1, and the curing time was 1–2 days. The final thickness of the laminates was about 6 mm. Finally, the laminates (stitched CFRP laminate as shown in Figure 1c) were cut into standard impact specimens with a length of 150 mm and a width of 100 mm by the CNC water-jet cutter. A total of five groups of different types of specimens were prepared for this experiment, as shown in Table 1.

### 2.2. Edge Impact Tests

In the study, the edge impact test was performed on a drop-weight impact tester, the CEAST 9340, with reference to ASTM D7136 [29], a standard for testing the damage resistance of composite laminates, as shown in Figure 2a. An edge impact fixture that can be used for edge impact experiments on composite laminates was designed by modifying the existing impact fixture in the laboratory, as shown in Figure 2b. The edge impact fixture was designed to adapt to the sample size so that the impact point was centered exactly on the long side of the laminates. To avoid the difficulty of impact positioning, a wedge impactor was chosen for this experiment (Figure 2c). The total mass of the impactor was 5.5 kg. The impactor diameter was 16 mm. The impact point was in the middle of the long side, and the impact direction was the negative direction of the Y axis. Because its length is greater than the specimen thickness, it could be in full contact with the side edge of the laminates in the thickness direction. The drop height of the impactor was automatically adjusted according to the set energy. Impact energy levels of 5, 10 and 15 J were used to conduct edge impact experiments on stitched and unstitched CFRP laminates. The impact experiments were repeated at least three times for each energy level in this study. The impact response curves were recorded by data acquisition system DAS 64 K-SC connected to a PC.

### 2.3. Compression after Edge Impact Test

The compression after the edge impact test was carried out on the ETM105D universal testing machine according to ASTM D7137 [30]. To prevent damage to the specimens caused by edge impact from damaging during the compression clamping process, four 5 mm thick aluminum pads were placed between the laminate and the compression fixture, as shown in Figure 3, and the compression loading speed was 1.25 mm/min.

## 3. Results Analysis of Impact Response

### 3.1. Effect of Stitching on Edge Impact of CFRP Laminates

Figure 4 shows the edge impact response results of unstitched CFRP laminates group A, stitched CFRP laminates group B (stitch density 10 mm × 10 mm) and stitched CFRP laminates group C (stitch density 15 mm × 15 mm) at impact energy levels of 5 J, 10 J and 15 J, respectively. The specimens of groups A, B and C have the same stacking sequence, P_1_, shown in Table 1. It is shown in Figure 4a,c,e that at the impact energy of 5 and 10 J, the impact force-time curves of the stitched and unstitched CFRP laminates have the same trend and the impact forces both show a rapid rise to a peak and then fall back to a certain height with the increase in response time, followed by a period of oscillation up and down near this height plateau, and then gradually a decrease to zero. However, when the impact energy is up to 15 J, the changing trend of the curve of impact force-time of the laminates is no longer the same. The impact force of the stitched and unstitched CFRP laminates at the impact energy of 15 J shows a continuous shaking upward until it rises to the peak force, and then the curves start to decline gradually. Based on the phenomena observed during the impact test, it can be explained that the fiber fracture and matrix cracking in a laminate may occur immediately while the impactor contacts the laminate when the impact energy is up to a certain level, and then a dent is generated at the impact point, in which many matrix fragments are accumulated. When the impactor continues to apply pressure on the laminates, which is similar to the crushing process of the composite matrix fragments [31], it results in an oscillating rising region in the curves shown in Figure 4c. However, the impact energy of 10 J and below may not be enough to cause matrix cracking in the laminates immediately, and failures such as matrix cracking and fiber breakage may not occur only until the impact energy is transferred to the interior of the laminates.

Figure 4b,d,f show the edge impact force-displacement curves of groups A, B and C CFRP laminates at the impact energy of 5, 10 and 15 J. The changing trend of impact force-displacement curves of CFRP laminates is the same as that of impact force-time curves of CFRP laminates shown in Figure 4a,c,e.

The peak impact force and maximum impact displacement of groups A, B and C CFRP laminates at the impact energy of 5, 10 and 15 J are shown in Figure 5.

The peak impact force refers to the maximum force that can be borne by the laminates at a certain impact energy, and the maximum impact displacement refers to the maximum deformation can be caused by the laminates at a certain impact energy. Figure 5a shows that at the same impact energy, the impact peak force of the stitched laminates is always greater than that of the unstitched laminates. Compared with the unstitched structure in group A, the average impact peak force of the stitched laminates in group C increases by 5.61%, 8.38% and 12.43% at the impact energy of 5 J, 10 J and 15 J, respectively, while the average maximum impact displacement decreases by 11.2%, 4.1% and 10.7% at the impact energy of 5 J, 10 J and 15 J, respectively. The peak impact average force of the stitched laminate in group B increases by 8.1%, 24.6% and 31.4%, and the maximum impact average displacement decreases by 12.5%, 15.6% and 21.9% at the impact energy of 5 J, 10 J and 15 J, respectively. Stitching can not only increase the maximum impact force of the laminates but also significantly reduce the maximum impact displacement of the laminates. This is because the integrity and rigidity of the composite laminates can be improved after the introduction of stitches, thus reducing the deformation of the laminates at the same impact energy.

With the increase in impact energy, the effect of the stitching of the CFRP laminates on the decrease in the maximum impact displacement and the increase in the peak impact force has an obvious increasing trend. This shows that the addition of the stitching process is conducive to improving the edge impact performance of CFRP laminates. The stitched laminates can bear more edge impact force than the unstitched laminates at the same impact energy, and the smaller deformation of the stitched laminates lead to smaller damage. After the impact force of the laminates reaches the peak, the force of the stitched laminates drops faster than that of the unstitched laminates, as shown in Figure 4e. It is more evident when the impact energy is higher than 10 J. This is because the introduction of the stitches also causes slight damage of the composite laminates in the stitching process. The stitch resin columns are not only damaged by impact load and can also inhibit the expansion of delamination during the impact process. The energy absorption and buffering capacity of the stitched laminates are therefore weakened. The higher the impact energy, the stronger the effect of the stitching becomes in the energy ranges used in the experiment above.

Compared the edge impact response results of group B and group C laminates with different stitching densities in Figure 4 and Figure 5, it appears that at the same impact energy, the average peak impact force of group B stitched CFRP laminates is always larger than that of group C stitched CFRP laminates, and the average maximum impact displacement of group B is always smaller than that of group C. It shows that the edge impact resistances of stitch density 10 mm × 10 mm stitched CFRP laminates are better than those of stitch density 15 mm × 15 mm stitched CFRP laminates. The reason is that the number of internal stitch resin columns in the group C laminates is less than that in the group B laminates, and the increased stiffness of group C laminates is smaller than that of the group B laminates. The effect of inhibiting the interlaminar crack propagation of the group C laminates by edge impact is worse than that of the group B laminates, resulting in weaker impact resistance of the group C laminates. Therefore, it can be concluded that the denser the stitch density, the stronger the edge impact resistance of CFRP laminates. However, this conclusion is limited to the two stitch densities involved in this experimental study. With the continuous increase in the stitch density, whether the edge impact performance of CFRP laminates will continue to increase needs to be further studied by adding more experiments.

### 3.2. Effect of Different Stacking Sequences on Edge Impact of CFRP Laminates

Figure 6 shows the edge impact response results of CFRP laminates in groups C, D and E at the impact energy levels of 5 J, 10 J, and 15 J, respectively. The corresponding stacking sequences of groups C, D and E are P_1_, P_2_ and P_3_, respectively, with the same stitch density of 15 mm × 15 mm, as shown in Table 1.

First, as shown in Figure 6a,c,e, the order of the peak impact forces for the specimens with different stacking sequences at the same impact energy is: P_3_ > P_1_ > P_2_. Group E with the stacking sequence P_3_ has the biggest peak impact force.

The peak impact force of the group D laminates is lower than that of the group C laminates, probably because their middle plies are laid with 0° fibers, which are perpendicular to the direction of edge impact. When the impactor falls and hits the laminates, the first fracture should occur in the 0° fiber plies. Therefore, during the edge impact experiments, when the middle plies of group D laminates with stacking sequence P_2_ break, an obvious dent will occur immediately below the impact point, followed by rapid propagation of the impact crack, and the laminates will soon be unable to continue to withstand the greater compressive stress of the impactor, and the peak impact force of the group D laminates will be minimal.

Figure 6b,d,f show the edge impact force-displacement curves obtained from the laminates of groups C, D and E at the impact energy of 5, 10 and 15 J. Figure 7 shows the average peak impact force and maximum impact displacement of the group C, D and E specimens. In Figure 7b, it is easy to find that the order of the maximum impact displacement of the laminates of groups C, D and E is D > C > E. Group D laminates with stacking sequence P_2_ have the worst impact resistance. Because they suffer the most severe plastic deformation after the impact, their maximum impact displacements are the largest compared to the other two groups of laminates. Group E laminates with stacking sequence P_3_ own the best impact resistance and the smallest plastic deformation, the maximum and final displacements after edge impact among the three groups of specimens. This will be illustrated in Figure 8.

### 3.3. Impact Damage Detection

#### 3.3.1. Optical Observation

Optical observation is the most common damage detection method of composite aircraft structures after impact, through which obvious damage such as impact dents, fiber breaks and delamination cracks can be found. Visual observation is both convenient and quick. The observed depth of dents on the impact surface of the laminates, the number of fiber breaks and the length of delamination cracks can be a preliminary judgment basis of the damage degree of structures. In the paper, visual observation was used to examine the damage of the laminates after the edge impact experiments, and to preliminarily compare the damage severity between different specimens.

Figure 8 shows the damage morphology of each group of laminates after edge impact at impact energy 5 J, 10 J and 15 J, respectively. From the damage shown in Figure 8, with the same stacking sequence laminations (groups A, B and C), compared with stitch density 10 mm × 10 mm (group B) and stitch density 15 mm × 15 mm (group C) CFRP laminates, the delamination induced by edge impact in the unstitched CFRP laminates (group A) is the most serious, such that the delamination crack length and crack opening are significantly largest at the same impact energy. This shows that the existence of stitches can improve the rigidity of the composite laminates on the one hand and inhibit the propagation of delamination cracks on the other. The delamination crack length of group B CFRP laminates is significantly shorter than that of the group C laminates at the same impact energy. The dents area of the group C CFRP laminates is larger than that of the group B laminates, which indicates that the edge impact resistance of the group B CFRP laminates with stitch density 10 mm × 10 mm is stronger than that of the group C CFRP laminate with stitch density 15 mm × 15 mm. This shows that the denser the stitch density, the greater the effect of inhibiting the propagation of delamination cracks. In addition, Figure 8 shows that the group B CFRP laminate at 10 J energy impact causes a stitch fracture because the impact point just falls on the stitch, and the fracture of the stitch indicates that it can help to support the part of the impact energy.

With different stacking sequences (groups C, D and E), the damage of the laminates induced by 5 J energy impact is so minor that it is difficult to visually compare the damage degree by observed dents at the impact point. At the impact of 10 J energy, there is also no noticeable difference in the damage of the CFRP laminates. However, at the impact of 15 J energy, the damage of the CFRP laminates with stacking sequence P_3_ is significantly smaller than that of the other two stacking sequences of laminates, and the length of the delamination crack is significantly shorter than that of the other two groups of laminates. In addition, the damages of fiber cracks, matrix fracture and the debonding between fiber and resin matrix in group E are less than those in group C and group D; that is, the CFRP laminates with stacking sequence P_3_ have the best edge impact resistance performance among the three stacking sequences laminates. These phenomena are consistent with the conclusion drawn from the impact response curves of the laminates shown in Figure 6.

To study the delamination details of the laminates, an optical microscope was used to observe the delamination cracks of each laminate, as shown in Figure 8b–e. With the same stacking sequence P_1_ (groups A, B and C), we found that the main delamination cracks, shown in Figure 8a–c, are all located between the second and third ply from the surface; that is, the fiber plies are 0/90°. The average longest crack lengths of the impact surface is 66 mm in group A, 31 mm in group B and 48 mm in group C at an impact energy of 15 J.

Figure 8d indicates that there are many delamination points with the stacking sequence P_2_. The main cracks are ones distributed between the second and third ply from the surface; that is, the fiber plies are 90/0°. There are also delamination cracks distributed between the fourth and fifth fiber plies −45/0° and the eighth and ninth fiber plies 0/−45°. As seen in Figure 8e, with the stacking sequence P_3_, the prominent delamination cracks are distributed between the seventh and eighth fiber plies 0/90°, and the long cracks are distributed between the fourth and fifth fiber plies 45/0° from the surface. Therefore, the delamination cracks of the laminates after edge impact mainly propagate along the 0° direction, especially when the fiber plies are 90/0°.

#### 3.3.2. Micro-Computed Tomography

The basic principle of micro-CT is to perform cross-sectional tomography in the specimen according to the reflection or transmission law after the X-rays are emitted on the specimen surfaces, so as to obtain the two-dimensional cross-sectional image inside the specimen. In this study, the micro-CT imaging platform was used to perform tomography on the specimens after edge impact. Before CT scanning, a specimen should be fixed on a transparent PMMA clamp with adhesive tape, and then the clamp and specimen should be fixed on the test platform in the CT equipment by screws. During CT scanning, the specimen rotates 360° clockwise with the clamp to ensure that all surfaces of the specimen can be scanned. The scanning time of each specimen is about 5 min. After the scanning is completed, multiple images of the specimen on three cross-sections of XY, YZ and XZ (the direction of the X, Y and Z coordinates is shown in Figure 9) can be obtained, respectively.

Because the density at the position of the laminates changes after damage, and the light intensity changes with the different density of materials, so the damage of the specimen is characterized by the gray scale of the image in the CT scanning results. Figure 10 shows the damage in the XY, YZ and XZ cross-sections of CFRP laminates with the unstitched laminates in group A subjected to edge impact of 5 and 15 J, and the stitched laminates in group D subjected to edge impact of 15 J.

From the XY cross-sectional view, the specimen has an approximate semi-elliptical damaged area below the impact point because of the fibers cracks and resin matrix fracture induced by huge compression force from the impactor, while the 0 and ±45° fibers far from the impact point are subjected to the tensile stress of the laminate itself. From the YZ cross-section of the specimen, there is an approximate “V” shaped damaged area at the top of the specimen, and the thickness of the laminates below the impact point has been slightly expanded. From the XZ cross-sectional view of the specimen, a lot of delamination damage occurs in the laminates, and the delamination cracks extend along the longitudinal direction of the 0° fiber ply from the impact point; especially the delamination damage near the impact surface of the laminates is most serious. That is, the length and width of the delamination cracks are the largest. At the same time, the fibers and resin in the dents are squeezed outward in the thickness direction because of the impact, resulting in an increase in thickness near the impact point.

Compared with Figure 10a,b, with the increase in impact energy, the area and depth of semi-elliptical dents in the laminates also increase, and the delamination damage is more serious. At the same energy impact, compared with the unstitched laminates shown in Figure 10b, the stitched laminates shown in Figure 10c have significantly less semi-elliptical dents area, depth and delamination damage, which also shows that stitching can inhibit the propagation of delamination cracks.

The CT technique allows not only the observation of the damage of the specimen in each cross-section, but also the specific observation of the damage evolution along a certain coordinate direction. Figure 11 shows the damage evolution images of the XZ cross-sections of each group of specimens at the impact energy 5 and 15 J. In each graph, because CT scans layer by layer, the first layer is the impact surface and the corresponding XZ cross-section is taken along the impact direction—that is, the negative direction of the Y axis (as shown in Figure 2c). The scan spacing is 0.129 mm (which is determined by the scanning resolution of the equipment). For example, the nth layer has gone down by (n−1) spacing from the impact surface along the negative direction of the Y axis, which is 0.129 × (n−1) mm. The number of layers of the last image in each group of figures is the deepest layer where the damage reaches. For example, in Figure 11f, the deepest damage reaches 0.129 × (62−1) = 7.869 mm from the impact surface.

From each group of XZ cross-sections of the laminates, the damage of the specimen gradually decreases from the surface of the laminates to the inside and extends to a certain depth where no obvious damage can be observed from the CT scan. At the same time, the size of the dents, the length and the width of the delamination cracks also show a gradual decrease from the surface of the laminate to the inside. This is because the potential gravitational energy of the impactor is converted into kinetic energy when it impacts the surface of the laminates. Composite materials absorb the energy brought by impact through deformation, and the energy will decay continuously until it is zero, when it is transferred to the depth of materials.

Table 2 is the damage depth of each group at the impact energy 5 and 15 J, calculated according to Figure 11. Figure 11 and Table 2 show that the damage of composite laminates, such as the delamination cracks length, width and crack propagation depth, in the same group is more serious with the increase in impact energy. At the impact energy of 5 J, the damage depth of each group is almost the same because damage after the edge impact is small in each group. However, when impact energy is up to 15 J, the damage of each group increases. The damage of stitched laminates is smaller than that of unstitched laminates. This indicates that the existence of stitch resin columns can inhibit crack propagation in the laminates.

Moreover, combined with Figure 8 and Figure 11, it can also be found that with the stacking sequences P_1_ and P_2_, whether the stitched CFRP laminates (group B, C and D) or unstitched laminates, the delamination between the second ply and third ply near the impact surface of the laminates is the most serious—that is, the length and width of the delamination cracks are the largest, while the damage mode of the stacking sequence P_3_ (group E) is different from that of the stacking sequences P_1_ and P_2_.

Combined with Figure 8a and Figure 11b,c,g,h, the delamination crack length of the group B laminates with stitch density 10 mm × 10 mm is much shorter compared with the group C laminates with stitch density 15 mm × 15 mm with the same stacking sequence P_1_, while the depth of damage is almost the same. The influence of the stitch in the X direction is greater than that in the Y direction in the study.

With the same stitch density, whether at an impact energy of 5 or 15 J, the length of the delamination crack of the group E laminates with stacking sequence P_3_ is smaller than that of group C with stacking sequence P_1_ and group D with stacking sequence P_2_. The depth of the delamination cracks of the group E laminates is almost same as that of group C and smaller than that of group D. It was interesting to find that the dent depth of group D is smallest, but the depths and lengths of their delamination cracks are the greatest at the impact energy of 15 J, as shown in Figure 8a and Figure 11h–j, because of the stacking sequence with the existence of the second/third fiber plies 90/0° and the fourth/fifth fiber plies −45/0° from the surface.

## 4. Results Analysis of Compression after Edge Impact

### 4.1. Compressive Load and Displacement

Figure 12 shows the compressive load-displacement curves of CAEI of the CFRP laminates in groups A, B and C after the impact of 0 J (not-impacted), 5 J, 10 J and 15 J, respectively.

With the increase in compressive displacement, the load changes of the stitched and unstitched laminates are almost the same: they rise rapidly and almost linearly to the peak compressive load and then drop sharply, and the higher the impact energy, the smaller the peak compressive load and stiffness of the laminates.

The loss of laminate stiffness increases with the increase in impact energy compared with the stiffness of not-impacted laminates. The loss of laminate stiffness is very small at impact energy 5 J, and the loss of laminate stiffness is the largest at impact energy 15 J. This is due to the more serious damage inside the laminates at impact energy 15 J, which can be seen in Figure 8 and Figure 11.

The difference is that the initial slope of the compressive load-displacement curve of the stitched laminates is significantly lower than that of the unstitched laminates, which may be due to the fact that the addition of the stitch resin columns blocks the transfer of the fiber compressive stress inside the laminate. There are also some cracks in the stitch resin columns during the impact process, which leads to a linear growth trend of the compressive load of the stitched CFRP laminates only after a certain displacement buffer. Compared with stitched laminates, at the impact energy of 5, 10 and 15 J the compressive load of unstitched laminates decreases significantly compared with the compressive load of the corresponding not-impacted laminates. This is because, as shown in Figure 8 and Figure 11, the greatest damage of unstitched laminates (group A) occurs at the same impact energy.

Figure 12b,c shows compressive load-displacement curves of the CAEI of CFRP laminates with stitch density 10 mm × 10 mm of group B and stitch density 15 mm × 15 mm of group C at 0, 5, 10 and 15 J impact energy. The displacement difference between the two kinds of stitch density laminates during CAEI is not large, but the maximum compressive load of group B laminates with stitch density 10 mm × 10 mm is obviously higher than that of the group C laminates with stitch density 15 mm × 15 mm. This may be due to the fact that the number of stitch resin columns in the laminates increases when the stitch density increases. More stitch resin columns enhance the longitudinal compressive strength of the laminates, which can help the laminates to withstand greater compression load. Figure 11 also shows that the damage of group B laminates is smaller than that of group C at the same impact energy.

However, the existence of stitch resin columns will also slightly damage the integrity of the laminates, so increasing the stitch density cannot reduce the compressive failure displacement of the laminates. The maximum displacement of CFRP laminates with two stitch densities in the study during compression is similar.

### 4.2. The Compression after Edge Impact Failure Mode

It appears that only the compressive failure modes of stitched and unstitched laminates can be clearly distinguished by observing the CAEI failure modes of CFRP laminates. Figure 13 shows the CAEI failure mode of the unstitched laminates in Group A and stitched laminates in group B.

As shown in Figure 13, after the unstitched specimens are crushed, a long crack perpendicular to the compression loading direction appears in the center of the panel, extending from the impact side to the other side. There is a slight bending of the unstitched laminates. While the stitched laminates show overall buckling, the cracks on the laminate extend along the stitch drop and do not extend to the sides of the laminates. Therefore, it is concluded that the main damage mode of compression after edge impact for the unstitched CFRP laminates is delamination and that for the stitched CFRP laminates is overall buckling.

### 4.3. Analysis of Residual Compressive Strength

The residual compressive strength of each group of stitched and unstitched specimens after edge impact is calculated according to Equation (1) and the curves are drawn as shown in Figure 14.
(1)$$P_{\mathrm{CAEI}}=\frac{F_{\max}}{S}$$
where *P_CAEI_* is the residual compressive strength of the specimen, MPa; *F*_max_ is the maximum compressive force of the specimen, N; *S* is the compressive contact area of the specimen; and *S* = b × h (b is the width of the specimen and h is the thickness of the specimen), mm^2^.

At the same impact energy, the effect of stitches on residual compressive properties after edge impact can be expressed as ((the average residual strength of stitched laminates−the average residual strength of unstitched laminates)/the average residual strength of unstitched laminates) × 100%.

With the same stacking sequence, the residual compressive strength of A, B and C groups of laminates changes with impact energy as shown in Figure 14a. Whether stitched or unstitched CFRP laminates, compared with corresponding not-impacted laminates, their residual compressive strength after edge impact will be reduced, but the degree of reduction will be different. At the impact energy of 5, 10 and 15 J, the average residual compressive strength of unstitched CFRP laminates in Group A decreases by 22.9%, 31.5% and 41.3%, respectively. The average residual compressive strength of the group B CFRP laminates stitch density 10 mm × 10 mm decreases by 16%, 23.7% and 34.8%, respectively. The average residual compressive strength of the group C CFRP laminates with stitch density 15 mm × 15 mm decreases by 24.1%, 26.1% and 33.5%, respectively. With the increase in impact energy, the average residual compressive strength of each group decreases. This is because the damage caused by edge impact increases with the increase in impact energy, which can be seen in Figure 8, Figure 10 and Figure 11. In addition, at the same impact energy, compared with unstitched laminates, the damage of stitched laminates is smaller, as shown in Figure 10b,c, and the decrease in the average residual compressive strength is also smaller. Obviously, this is because stitching can inhibit damage propagation when the laminates are subjected to impact.

Compared with unstitched laminates, at 5, 10 and 15 J impact energy, the average residual compressive strength of the group C CFRP laminates with stitch density 15 mm × 15 mm increases by 5%, 15% and 20.9%, respectively. The average residual compressive strength of the group B CFRP laminates with stitch density 10 mm × 10 mm increases by 24.2%, 27% and 26.3%, respectively.

This is because the composite laminates have different damages after edge impact, which are shown in Figure 8, Figure 10 and Figure 11. The damage, especially the size and depth of dents and delamination damage, of unstitched laminates (group A) is the largest with the same stacking sequence and the same impact energy, which leads to the minimum residual compressive strength. With the increase in stitch density, the size and depth of dents and delamination damage of group B are smaller than those of group C at the same impact energy, which can be shown in Figure 8 and Figure 11, so the residual compressive strength of group B is higher than that of group C. Therefore, the properties of edge impact and compression after edge impact of stitched laminates are better than those of unstitched laminates. Among the two stitch densities involved in this experiment, the denser the stitch density, the stronger the properties of the edge impact and compression after edge impact of laminates.

With the same stitch density, the change of residual compressive strength of the C, D and E groups of laminates with impact energy is shown in Figure 14b. Compared with the corresponding not-impacted laminates at the impact energy of 5, 10 and 15 J, the average residual compressive strength of the group C CFRP laminates with stacking sequence P_1_ decreases by 24.1%, 26.1% and 33.5%, respectively. The average residual compressive strength of group D CFRP laminates with stacking sequence P_2_ decreases by 23.8%, 23.9% and 31.5%, respectively. The average residual compressive strength of group E CFRP laminates with stacking sequence P_3_ decreases by 2.9%, 4% and 15.6%, respectively. Obviously, at the same impact energy, the reduction in the average residual compressive strength of the group E CFRP laminates with stacking sequence P3 is the smallest. This is because the damage of group E laminates is the smallest among the three, which can be seen in Figure 8, Figure 10 and Figure 11. Therefore, the decrease in the residual compressive strength of group E is the smallest, and the decrease in the average residual compressive strength of group C and group D is similar because the damage caused by the edge impact at the same energy is also similar.

## 5. Conclusions

Damage of CFRP laminates is caused by edge impact, leading to reducing their residual compressive strength. Based on the above research, the following conclusions can be drawn:(1)The improvement effect of the stitching is more significant with the increase in the impact energy that is used in this study. Compared with the unstitched laminates with the same stacking sequence, the peak impact force of the stitched CFRP laminates with stitch density 15 mm × 15 mm has an increase of 4.35–12.43%. The peak impact force of the stitched CFRP laminates with stitch density 10 mm × 10 mm has an increase of 8.1–31.4%.(2)The damage of edge impact of stitched laminates is smaller than that of unstitched laminates at the same impact energy. The main failure mode of compression after edge impact of unstitched CFRP laminates is delamination and that of stitched CFRP laminates is global buckling.(3)At the impact energy of 5, 10 and 15 J, the average residual compressive strength of unstitched CFRP laminates in group A decreases by 22.9%, 31.5% and 41.3%, respectively. The average residual compressive strength of group B CFRP laminates with stitch density 10 mm × 10 mm decreases by 16%, 23.7% and 34.8%, respectively. The average residual compressive strength of group C CFRP laminates with stitch density 15 mm × 15 mm decreases by 24.1%, 26.1% and 33.5%, respectively.(4)With the increase in stitch density and impact energy, the influence of stitching on the residual compressive strength after edge impact of composite laminates increases significantly. At 5, 10 and 15 J impact energy, compared with unstitched laminates, the average residual compressive strength of group C CFRP laminates with stitch density 15 mm × 15 mm increases by 5%, 15% and 20.9%, respectively; the average residual compressive strength of group B CFRP laminates with stitch density 10 mm × 10 mm increases by 24.2%, 27% and 26.3%, respectively.(5)Compared with corresponding not-impacted laminates, the average residual compressive strength of group E CFRP laminates with stacking sequence P_3_ decreases the least at the same impact energy. At the impact energy 5, 10 and 15 J, the average residual compressive strength of group C CFRP laminates with stacking sequence P_1_ decreases by 24.1%, 26.1% and 33.5%, respectively. The average residual compressive strength of group D CFRP laminates with stacking sequence P_2_ decreases by 23.8%, 23.9% and 31.5%, respectively. The average residual compressive strength of group E CFRP laminates with stacking sequence P_3_ decreases by 2.9%, 4% and 15.6%, respectively.

The P_3_ stacking sequence has the best impact resistance properties compared to the other two stacking sequences. One of the main reasons is that there may be the fiber plies ±5° in the laminates, which can decrease their impact damage at the same impact energy. In addition, the existence of the fiber plies ±45° may be insensitive to the damage, which can decrease the degradation of the residual compressive strength of the laminates. So, it can improve their properties of impact resistance and damage tolerance. The mechanism of excellent impact resistance of stacking sequence P3 needs to be further studied and revealed.

## Figures and Tables

**Figure 1 materials-15-08822-f001:**
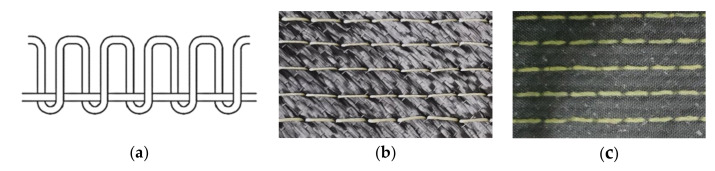
Manufacturing process of stitched CFRP laminate. (**a**) Modified locking stitch method; (**b**) stitched CFRP preform; (**c**) stitched CFRP laminate.

**Figure 2 materials-15-08822-f002:**
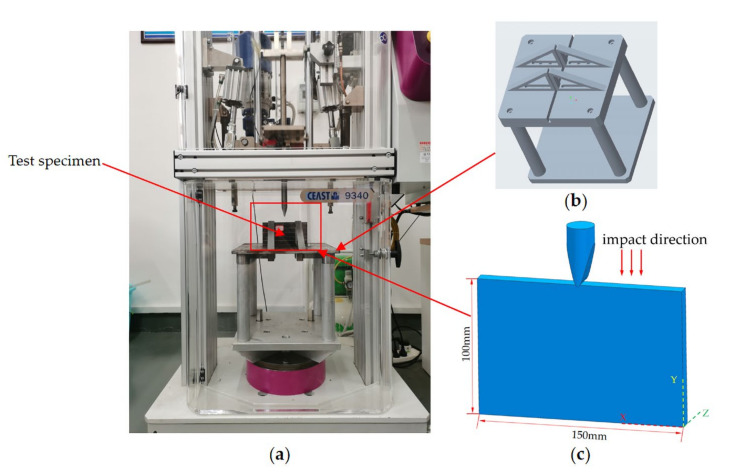
Edge impact test device. (**a**) Drop-weight impact machine; (**b**) edge impact fixture; (**c**) impactor and impact direction.

**Figure 3 materials-15-08822-f003:**
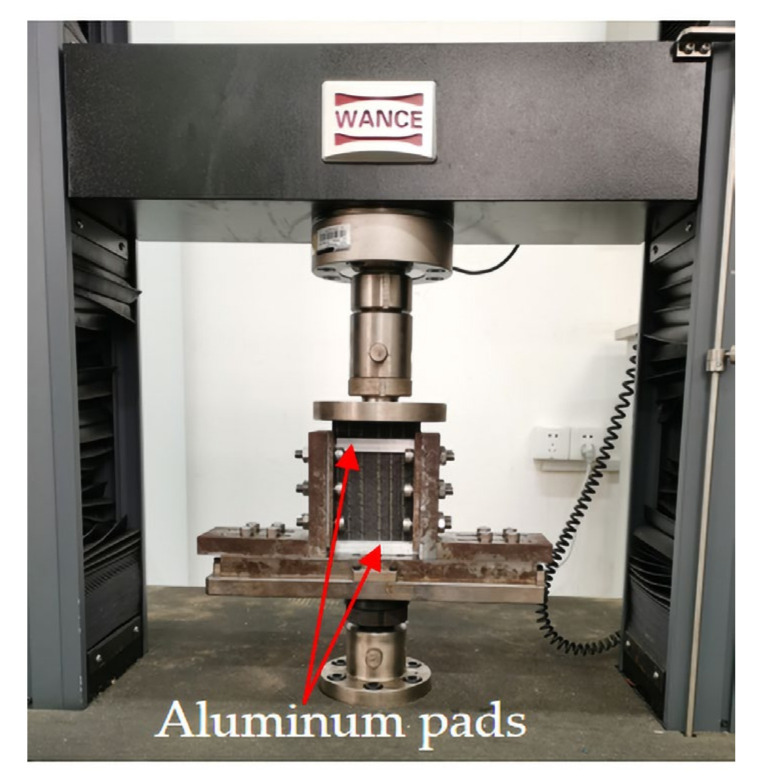
Compression after edge impact test device.

**Figure 4 materials-15-08822-f004:**
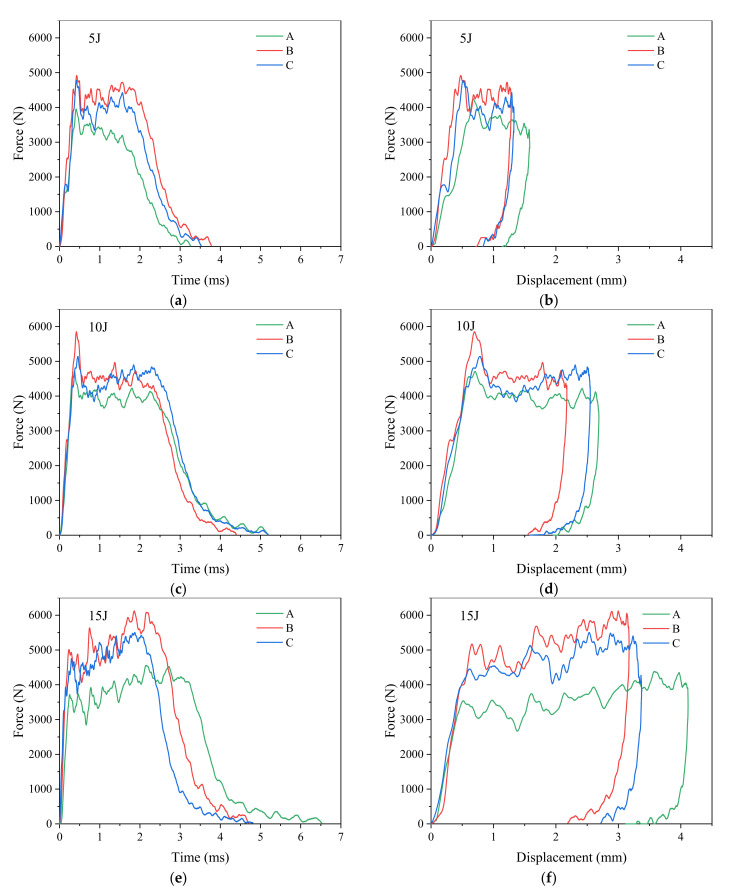
Edge impact response of groups A, B and C specimens at an impact energy of 5, 10 and 15 J. (**a**,**c**,**e**) Impact force-time curves; (**b**,**d**,**f**) impact force-displacement curves.

**Figure 5 materials-15-08822-f005:**
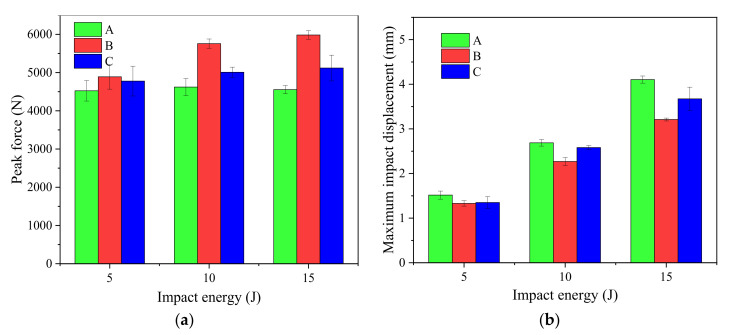
Peak impact force and maximum impact displacement of edge impact of specimens in groups A, B and C. (**a**) Peak impact force; (**b**) maximum impact displacement.

**Figure 6 materials-15-08822-f006:**
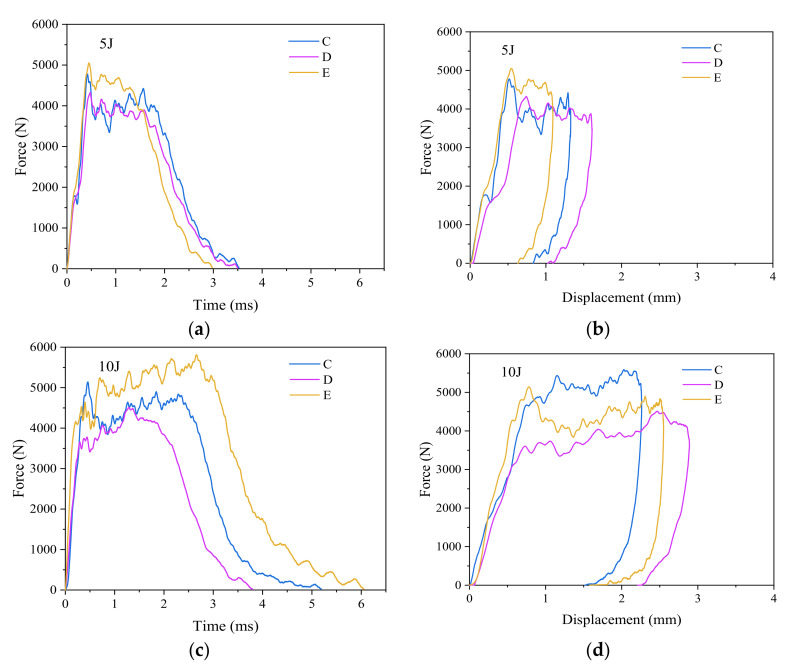
Edge impact response of groups C, D and E specimens at the impact energy of 5, 10 and 15 J. (**a**,**c**,**e**) Impact force-time curves; (**b**,**d**,**f**) Impact force-displacement curves.

**Figure 7 materials-15-08822-f007:**
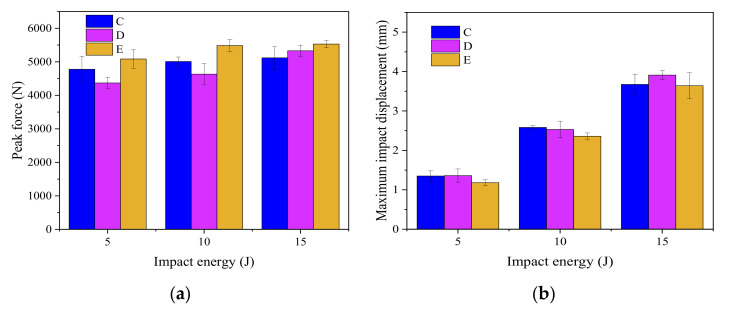
Peak impact force and maximum impact displacement of edge impact of specimens in groups C, D and E. (**a**) Peak impact force; (**b**) maximum impact displacement.

**Figure 8 materials-15-08822-f008:**
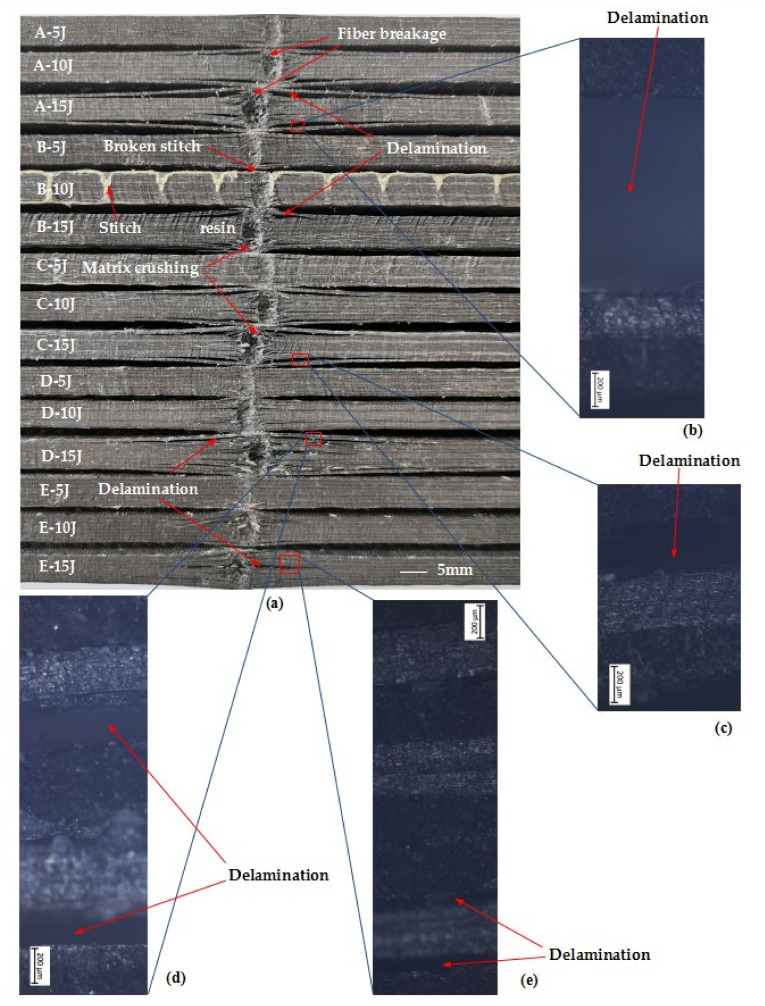
Damage morphology of each group laminates after edge impact at impact energy 5, 10 and 15 J. (**a**) Picture; (**b**–**e**) optical microscope picture.

**Figure 9 materials-15-08822-f009:**
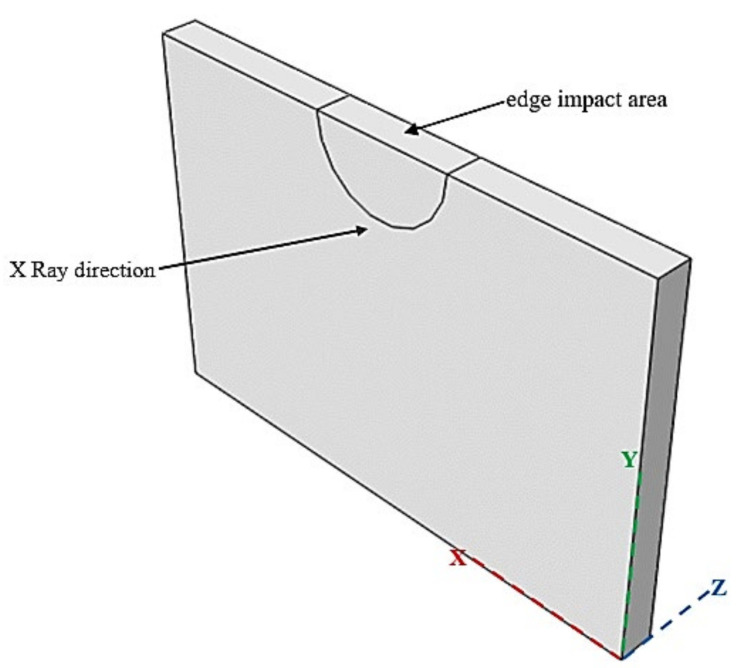
X-ray and specimen coordinate direction.

**Figure 10 materials-15-08822-f010:**
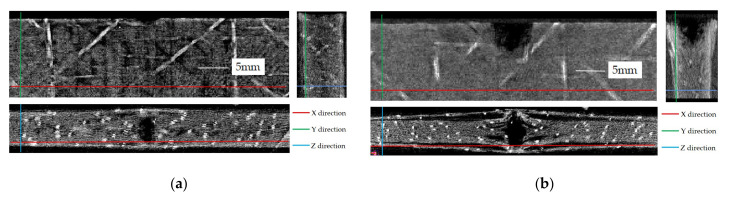
Micro- CT scan results of specimens after edge impact. (**a**) A-5 J; (**b**) A-15 J; (**c**) D-15 J.

**Figure 11 materials-15-08822-f011:**
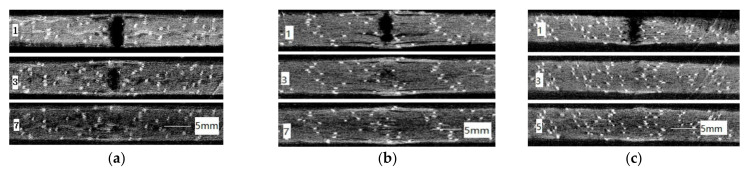
Damage evolution in XZ cross-sections of each group specimens after edge impact at the impact energy 5 and 15 J. (**a**) A-5 J; (**b**) B-5 J; (**c**) C-5 J; (**d**) D-5 J; (**e**) E-5 J; (**f**) A-15 J; (**g**) B-15 J; (**h**) C-15 J; (**i**) D-15 J; (**j**) E-15 J.

**Figure 12 materials-15-08822-f012:**
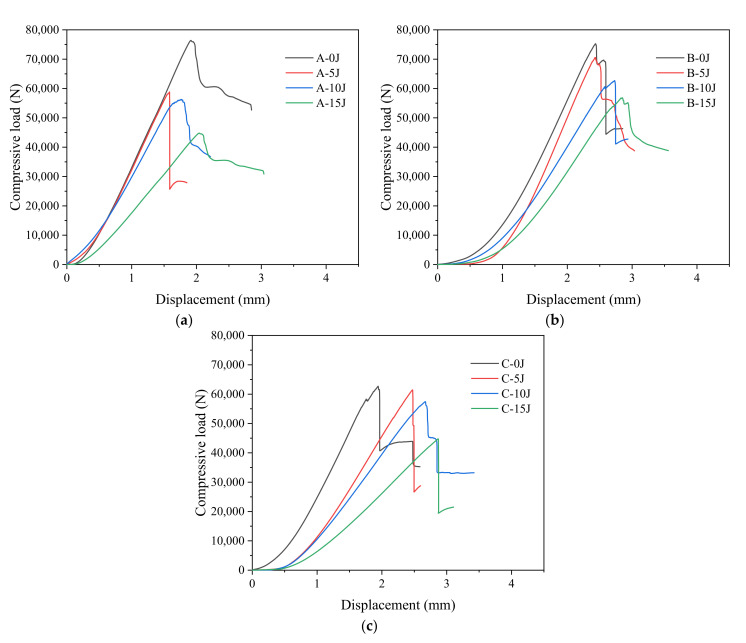
Compressive load-displacement curves of CAEI of groups A, B and C laminates. (**a**) Group A; (**b**) group B; (**c**) group C.

**Figure 13 materials-15-08822-f013:**
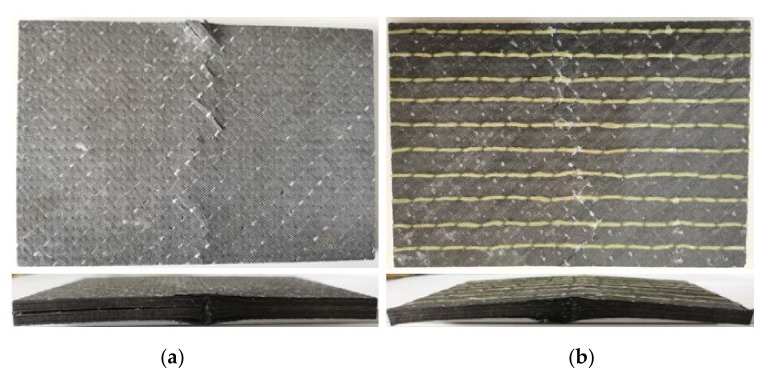
Failure mode of CAEI in the laminates. (**a**) Unstitched laminate; (**b**) stitched laminate.

**Figure 14 materials-15-08822-f014:**
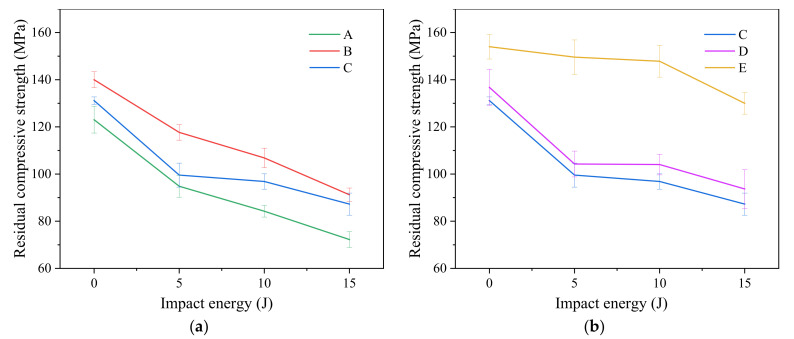
Residual compressive strength-impact energy curves of each group laminates. (**a**) Groups A, B and C; (**b**) groups C, D and E.

**Table 1 materials-15-08822-t001:** Types of specimens.

Specimen Group	Craft	Stitching Density/(mm)	Stacking Sequence Symbol	Stacking Sequence	Number of Layers	Impact Energy/(J)
A	Unstitched	/	P_1_	[−45/0/90/45/90]_2s_	20	0 (no impact), 5, 10, 15
B	Stitched	10 × 10	P_1_	[−45/0/90/45/90]_2s_	20	
C	Stitched	15 × 15	P_1_	[−45/0/90/45/90]_2s_	20	
D	Stitched	15 × 15	P_2_	[45/90/0/−45/0]_2s_	20	
E	Stitched	15 × 15	P_3_	[45/0/90/0/−45]_2s_	20	

**Table 2 materials-15-08822-t002:** Damage depth of each group at the impact energy levels of 5 and 15 J.

	Specimen Group	A	B	C	D	E
Damage Depth (mm)						
Impact Energy (J)						
5		0.774	0.774	0.645	0.516	0.516
15		7.869	3.096	3.225	6.708	3.612

## Data Availability

Not applicable.

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
