# Peer review of "Effect of Stitching, Stitch Density, Stacking Sequences on Low-Velocity Edge Impact and Compression after Edge Impact (CAEI) Behavior of Stitched CFRP Laminates"

_materials, 2022, doi:10.3390/ma15248822_

Round 1

Reviewer 1 Report

 Nowadays CFRP laminates are commonly used in aircraft industry, therefore analysis of CFRP disadvantages/weakness and methods of their improvement are important for reliability of aircraft structures.

 The paper presents interesting analysis of CAEI behaviour of stitched laminates. The methodology of investigation is appropriate. The results are presented clearly and they can be useful in engineering practice. However, a minor revision of the article is necessary.

A few comments and exemplary recommendations are listed below.

 Specimen preparation and experimental tests have been described at the beginning of the paper in section 2, however, stitch and impact directions are not clear until figure 12 (section 4). In my opinion specimen configuration (including stitch and impact directions) could therefore be presented in section 2.

 Line 34: “… because of their excellent properties such as light weight and high-strength

 72: “ … with two stitch densities … “

 Line: 96: Shape of wedge impactor especially diameter of impactor leading edge would be also interesting for the reader.

 Lines 96-97: “Because its length is greater than the specimen thickness, it could be …” ?

 Figure 2a, Figure 3: The image quality is poor.

 Figure 4, 6, 11: Grid lines could be shown in the chart for better readability.

 Table 2: Please consider presenting the results for C, D, E specimen groups in the same way as for A, B, C (like in Fig. 5) including deviations (error bars).

 Lines 213-214: The sentence is not clear.

 Figure 7: Please consider improving the quality of pictures in Figure 7 (b-e).

 Line 303: Please use degree instead of zero.

 353: “In each graph …“

 Lines 360-361: Please consider comparing the depth of damage profiles in the table.

 Line 533: “5. Conclusions”

 535: “… leading to reducing … “

Author Response

Dear Review:

On behalf of my co-authors, we are very grateful to you for giving us an opportunity to revise our manuscript.  I have made in response to the your questions and suggestions on an item-by-item basis. Please see the attached file. 

Thank you and best regards.

Yours sincerely,

Jiamei Lai.

Reviewer 2 Report

The authors of the manuscript titled Effect of Stitching, Stitch Density, Stacking Sequences on Low-velocity Edge Impact and Compression after Edge Impact (CAEI) Behavior of Stitched CFRP Laminates report experimentally obtained effect of an edge impact (with various impact energies) on the compressive performance (namely, stiffness and strength) of the carbon-reinforced samples with and without stitches (in three stacking sequences). The authors demonstrate how significant impact the stitching has on the response.

The message of the manuscript is clear, the experiments are comprehensively described, and the results are exhaustively commented. However, the whole manuscript outline is a bit simplistic in my opinion and I am missing explanations behind some observations, e.g., I am missing at least a hypothesis why the P3 stacking sequence has the best resistance compared to the other sequences.

Even though I am not a native speaker (and I am fully aware of the fact), I think that the English can be improved as well. There are several formulations, in which I had difficulties following the text. Below, I list several examples (keep in mind that the list is far from exhaustive):

·         Line 132, p. 4: “When the impactor continues to apply pressure on the laminates, which is similar to the crushing process of the composite matrix fragments[31], resulting in an oscillating rising region as shown in the Figure 4(c).
The first sentence combined with “…, resulting” does not make sense to me.

·         Line 162, p. 5: “on the decrease of the maximum impact displacement and the increase of the peak impact force has an obvious increase trend.
increase -> increasing?

·         Line 334, p. 13: “of interlaminar delamination damage occurs in the laminates,”
Isn’t delamination always interlaminar?

·         Line 375, p. 14: “Comparing at the impact energy of 5J shown in Figure 10 (a-e), and at the impact 375 energy of 15J, as shown in Figure 10 (f-j), we can find that the damage of composite 376 laminates in the same group is more serious with the increase of impact energy, and the 377 delamination cracks length, width and crack propagation depth also increase.
Comparing what?

·         Line 534, p. 18: “CFRP laminates will cause damage after being subjected to edge impact…
I am rather sure that the laminates will not cause damage, but they will be damaged/experience damage.

At the same time, I believe that a better choice of tenses would improve readability of the manuscript. For instance, the description of the CT scanning (lines 310 and 311) reads: “Before CT scanning, a specimen shall be fixed on a transparent PMMA clamp by adhesive tape, and then the clamp and specimen shall be fixed on the test platform…”. Should or even was is a better choice in my opinion as it clearly states that this is what you did.

Personally, I also find the formulations such as “it can be seen that”, “it can be shown”, or “it can be found” too repetitive in the text, which consequently makes the reading a tedious task.

I am also missing a consistency in the use of spaces between numbers and unit, e.g., sometimes, the authors write 10J, other times 10 J. The same holds also for spaces after comma (e.g., “… group C, D and E“ vs. “… group C ,D and E“), hyphen in Micro-CT, space between figure number and the letter of a subfigure, and space between a number and the times sign.

A weak point of the manuscript is the low resolution of many pictures. Since many paragraphs refer to figures, I strongly recommend providing images with a (much) higher resolution.

Figures 5 and 13 show also confidence intervals for each value of the impact energy, indicating that there were several samples in one group. However, I didn’t find a mention on how many samples were tested for each group. If there were more samples, then I am missing the same statistical information in Figures 4, 6, and 11. In my opinion, it would be instrumental to see the scatter in Figs. 4, 6, and 11 as well.

I would also recommend reshuffling of graphs in Figure 4 such that the left column corresponds to force-time graphs while the second column presents force-displacement responses. Each row will then represen a particular impact energy.

Taking the above-mentioned into account, I cannot recommend the manuscript for publication, and I think it needs a major revision before it can be reconsidered.

Author Response

Dear Review:

On behalf of my co-authors, we are very grateful to you for giving us an opportunity to revise our manuscript.  I have made in response to  your questions and suggestions on an item-by-item basis. Please see the attached file. 

Thank you and best regards.

Yours sincerely,

Jiamei Lai

Reviewer 3 Report

In this study, the effects of stitching, stitch density, stacking sequence and impact energy were experimentally investigated in order to present the effects of stitching on the edge impact and CAEI properties of CFRP laminates.

The study is topical, original and technically good, and the conclusions are sound and clear. The manuscript is very well organized and since this study might be of interest to the structural engineering community I recommend this paper for publication.

It would only be helpful if some pictures were presented more clearly. For example; Fig. 2, Fig. 3, Fig. 9c and Fig. 10g.

Author Response

(The authors gave the same response as above.)

Reviewer 4 Report

Paper is well written and experimental testing is enough explained. Please follow all the suggestions given in the attached file. Here are mentioned the most important ones:

- leave space before references and before units of measurement;

- too many references cited as [18-28]; write explanations on them in separate categories;

- in Figs. 2 and 3 improve picture resolution;

- in Fig. 4 choose not so close colors for curves A and C; they are confusing the reader as being similar - choose one of them as being black or some other color;

- in Fig. 6 choose not so close colors for curves E and C; they are confusing the reader as being similar - choose one of them as being black or some other color;

- in Fig. 7 change pictures (b)-(e); now they are not clear;

- in Fig. 11 make curves for 0 J with brack color;

- in Fig. 13 A and C should be visibly; use different colors

- in Figs. 11 why do you write units "/N" and nor " (N)"? and why why "/mm" and not " (mm)"?

- in Fig. 13 same for MPa and J; A and C should be visibly different colors;

- Conclusions are too long and difficult to follow; write them in a brief manner.

Author Response

(The authors gave the same response as above.)

Round 2

Reviewer 2 Report

Most comments from my previous review have been addressed. I still believe that, e.g.,:

* “CFRP laminates cause damage after edge impact” is an unfortunate formulation and the laminates do not cause any damage, but the damage is caused by the impact to them;

* “When the impactor continues to apply pressure on the laminates, which is similar to the crushing process of the composite matrix fragments[31], resulting in an oscillating rising region in the curves shown in the Figure 4c” is not a complete sentence, because it starts with when but there is no other sentence after it. I suggest replacing resulting with it results

However, these are only stylistic recommendations.

It is a pity that the authors do not provide any hypothesis why the response of P3 stacking is superior. Simply stating the difference from the other stacking sequences does not explain anything. But perhaps, as the authors state in the last part of the manuscript, they will investigate this effect in their subsequent work.

Author Response

We are very grateful to you for giving us an opportunity to revise our manuscript again. We try our best to revise our manuscript according to the comments. Please see the attached file for our responses on the comments and suggestions. Thanks again to your hard work.
